# Lab-Scale Methodology for New-Make Bourbon Whiskey Production

**DOI:** 10.3390/foods12030457

**Published:** 2023-01-18

**Authors:** Virginia L. Verges, Jarrad W. Gollihue, Glenna E. Joyce, Seth DeBolt

**Affiliations:** 1Department of Horticulture, University of Kentucky, Lexington, KY 40503, USA; 2Department of Biosystems and Agricultural Engineering, University of Kentucky, Lexington, KY 40503, USA; 3James B. Beam Institute for Kentucky Spirits, University of Kentucky, Lexington, KY 40503, USA

**Keywords:** bourbon, new-make whiskey, maize, fermentation, distillation, cereals

## Abstract

Whiskey production originated in Scotland in the 15th century and was based on malted barley. As Scotch-Irish settlers came into the Ohio river valley, they began fermenting and distilling the primary grain of North America, maize. These earlier settlers started a heritage; they created American Whiskey. The bourbon industry in Kentucky had tremendous growth in the last 20 years, and currently, distilleries have a broad increase in product innovation, new raw materials, improved sustainability, efficient processes, and product diversification. Our study presents a new lab-scale method for new-make bourbon whiskey production. It was developed to mimic distilleries’ processes; therefore, results can be extrapolated and adopted by commercial distilleries. The method focused on reproducibility with consistency from batch to batch when handled by an operator or small crew in a university lab. The method consisted of a first cooking step to make a “mash”, a fermentation phase of 96 h, a first distillation accomplished with a copper pot still to obtain the “low wines” and a second distillation carried out with an air still to collect the “hearts”. The method produced a final distillate of 500–700 mL for further sensory analysis and tasting. This lab-scale method showed consistency between samples in the different parameters quantified and will be also used to train students in fermentation and distillation studies.

## 1. Introduction

Bourbon whiskey is a class of whiskey that, by the Standards of Identity for Distilled Spirits (“Title 27, Part 5, Subpart C” of the US Code of Federal Regulations), must contain at least 51% maize (*Zea mays* L.), must be aged in new charred oak containers, must be distilled to no more than 160° proof, and must enter into the barrel for aging at not more than 125° proof. Besides corn (“common” name for maize in the US), bourbon distillers use other grains for cooking a mash, namely wheat and rye, the most commonly added in small quantities. Some bourbon whiskeys reach 80% corn grain in the total mash, with corn becoming the main source of fermentable sugars in bourbon. Bourbon is native to Kentucky and originates in the whiskey distilled by early Scotch-Irish frontier settlers in America [1]. The bourbon industry is a US 220 million industry in Kentucky [1,2,3] and has seen staggering growth since the early 2000s with the emergence of craft distilleries and a renewed popularity of the spirit.

Distillation has been carried out from the earliest times using pot stills that were heated directly by open fires in a furnace or hearth; currently, it is a requirement to distill some classes of whisky in copper pot stills; this process is called “batch distillation.” The other most common distillation is with a continuous still, which continuously distills whiskey in vertical hollow columns/towers with associated piping, heat exchangers, pumps, storage vessels, and support structures [4]. Batch or pot stills employ double or triple distillation and generate a highly flavored spirit; continuous stills provide lighter flavored spirits [5]. Bourbon whiskey distillation can be produced by a pot still, a column still, or the generally used method that is a combination of both, with a column still feeding into a doubler that functions similarly to a pot still. Doublers in pot still distillation can imply a secondary distillation step. They function as a pseudo-boiler, meaning the energy from the initial distillation provides the energy for the second distillation; in the case of bourbon, distillation doublers have an energy source. The doubler typically raises the alcohol content of the spirit by 5–8% ABV, and the end product or “high wine” is typically recovered at 65–80% ABV [6]. 

The emergence of new consumers due to the exceptional development of the craft distilling industry in the United States [7] has been impactful. Added to this are the increasing expectations of traditional consumers [8] eager to try new flavors and aromas. These factors together with the interest of the industry to use new grain bases [9] and increase the sustainability of bourbon production have resulted in a revitalized synergy between the industry and research centers and universities. Establishing a protocol that emulates the whiskey production process of mashing, fermentation, and distillation is crucial to have a methodology that can be robust enough to meet research goals. This method should be in line with the distilleries’ processes; therefore, results could be extrapolated and adopted by commercial distilleries but, at the same time, should be handled by an operator or small crew in a university lab. Any method should be reproducible among replicate samples and have clear time points to evaluate the treatments and later draw conclusions. Different methodologies to handle the whisky production process are found in the literature; many are for malt whisky production, which differs from grain whiskey production in the initial steps. Some research articles focus on evaluating alcohol yield and other parameters after fermentation and describe the methodology used to obtain both a mash and “distiller’s beer” (a common name for the fermented mash between 6 and 10% ABV) [10,11,12,13,14]. Some researchers report the results of the whole process from mashing to distillation [9,15,16]. A third methodology found is outsourcing the mashing and fermentation processes to a local distillery and focusing the research work on distillation parameters, flavor, and aroma [17,18].

Here we aimed to develop a new lab-scale methodology for new-make (i.e., unaged whiskey that is the immediate by-product of distillation) whiskey production simulating as much as possible a typical bourbon whiskey distillery operation. This lab-scale mashing, fermentation, and distillation methodology ensures consistency between samples reaching the accuracy needed to obtain reproducible and publishable results. We developed a method that meets the following goals: a final distillate of between 500 and 700 mL for further sensory analysis and tasting, does not require more than 3 kg of grain, and is flexible enough to allow for focusing on specific parts of the production process. Because it has been created in an effort to mimic the steps followed by distilleries, its results may be rapidly adopted or replicated on a bigger scale. To develop the lab-scale whisky production methodology, we took previously developed protocols found in the literature [5,12] as a reference, optimized them to meet the desired scale manageable in a lab by one operator, and evaluated the correct implementation of each part of the process through the quantification of sugars and ethanol.

## 2. Materials and Methods

### 2.1. Maize Samples

Eight maize (*Zea mays* L.) samples were used in this study. Each sample (2.5 kg) was obtained from different sources. One sample was provided by James B. Beam distillery (standard sample), and the other seven samples were provided by three agricultural operations in Kentucky: DK65-92 and P.1197 from Woodford County, KY, USA; P.1442 from Butler County, KY, USA; DK64-32 from irrigated and non-irrigated fields, from Marion County, KY, USA; and DK64-32 and DK63-58 from Nelson County, KY, USA.

### 2.2. Whole Kernel Assessment

Each whole corn kernel sample of 2.5 kg was sieved through a 5 mm round hand sieve to remove broken kernels and foreign material. Extra foreign material was removed manually. After grain cleaning, two subsamples of 300 g per sample were evaluated using a Perten Instrument DA7250 near-infrared (NIR) spectrometric determination grain analyzer (Stockholm, Sweden). The NIR determinations included moisture content (%), protein (%), crude fiber (%), ash content (%), starch content (%), and oil content (%). Sample 1 is the standard sample provided by James B. Beam distillery.

### 2.3. Mashing and Fermentation

After cleaning and NIR determinations, samples were milled with an electric blender to break, and then they were sieved with a sieve with circular holes of 2 mm diameter to avoid bigger kernel pieces remaining in the sample and ensure consistency from batch to batch. A steel cooking pot was filled with 7.8 L of tap water and set on top of an induction burner (Duxtop professional Portable Induction Cooktop, 1800 Watts, 120V). The water temperature was brought to 85 °C with the induction burner set to medium-high. When the water started to boil, 1 mL alpha-amylase (Distillazyme by Lallemand Biofuels & Distilled Spirits, Duluth, GA, USA) was added to the water and mixed. While the maximum enzyme activity occurs between 70 and 85 °C [19], we only saw a slight reduction in the enzyme activity when cooking the mash, and we observed the benefits of emulating a “pre-malting” process that makes the solution less viscous [20] and less likely to burn. Three minutes later, 2100 g of milled corn was added to the pot and manually mixed until uniform with no clumping (30 s mixing with a spoon). The temperature was maintained at 85–90 °C for 90 min. The mash was regularly mixed with a spoon to prevent burning, and when not mixing, the pot was covered with a lid to prevent excessive evaporation. After 90 min, the heat was turned off, and at 85 °C another 1 mL of alpha-amylase was added to the mash. The pot remained in the lab on a clean bench, cooling down at room temperature. The mash rested until the temperature reached 35–37 °C. At this point, 1 mL gluco-amylase (Distillazyme by Lallemand Biofuels & Distilled Spirits) was added, and the mash kept cooling to 32 °C. To estimate the amount of fermentable sugars, density in degree Brix (°Bx) was measured with a refractometer before adding the yeast. The target was approximately 18 °Bx before the onset of fermentation to achieve approximately 10% ABV in the final beer. If this density was achieved and the mash’s temperature was below 32 °C, the yeast solution was added to the mash. To start fermentation, 6 g of active dry *Saccharomyces cerevisiae* (“DistilaMax” by Lallemand Biofuels & Distilled Spirits) was added to 100 mL of 40 °C water along with 3 g yeast nutrient (“DistilaVite” by Lallemand Biofuels & Distilled Spirits) and 1 g of sucrose. This mixture was gently swirled for 5–10 s until dissolved. The yeast amount was based on a slight increase in the recommendation by Lallemand [21]. The DistilaMax strain of *Saccharomyces cerevisiae* was chosen for its neutral flavor that allows for focusing on the maize grain effect for further aroma and flavor analysis.

After a slow mixing to incorporate the yeast solution into the mash, the mash was transferred to a 7.6 L plastic fermenter previously sanitized with Star San (Five-star Chemical & Supply Inc., Arvada, CO, USA). The mash bucket was covered with a lid with a bubbler airlock. Fermentation proceeded for four days (96 h) in an incubator set at 30–32 °C. The decision to let the fermentation occur for four days was made after a trialing phase of 48 to 120 h. With a manual hydrometer (Brewer’s Elite Hydrometer for Beer and Wine), we measured the sugar level from day 2 to day 5 in different batches. After four days of fermentation, we achieved the desired fermentable sugar level (≤1.00 °Bx) in most batches. Therefore, to standardize the procedure, we decided to take day 4 (96 h) as the end of fermentation. After fermentation, the fermenter was taken from the incubator, sugars were measured with a manual hydrometer to confirm the density was reduced to <1.00 °Bx, and the mash was prepared to initiate distillation. 

### 2.4. First Pass and Second Pass Distillation 

The mash, now called “beer”, was added to a 9.5 L copper pot still. The copper pot still had a condenser filled with cool water that produced the ethanol condensation after being first evaporated by heating the “beer”. The copper pot still was placed on top of the induction burner for the distillation process to occur, and the temperature was set to 207 °C until the pot still thermometer reached 76 °C. At this point, the heat was reduced to 190 °C to avoid burning the “beer”, and the pot still reached 88 °C in 15–20 more minutes. The still thermometer should stay between 88 and 92 °C for optimal distillation to ensure the beer and grains do not burn. The distillation proceeded until the collection of 700 mL of distillate (“low wines”) was complete. The alcohol concentration by volume of the low wines was measured using an alcohol hydrometer, and the low wines were diluted by adding deionized (DI) water to the desired alcohol concentration (40% *v*/*v*) before starting the second distillation to keep uniformity and consistency from batch to batch. The pot still was cleaned first with a mild caustic solution to remove any stillage on the still wall, followed by a water rinse, then a mild citric acid solution, and another rinse, between batches to ensure no mix or contamination affected the process. For the second distillation, a 4 L stainless air still (Turbo Air Still by Still Spirits, Auckland, New Zealand) with an air fan-cooled condenser and electric heating was used. The diluted low wines (800–1000 mL) were added to the air still, and distillation commenced. The first 50 mL (“heads”) were collected in a glass jar; using a different glass jar, the following 300 mL were collected (“hearts”). The “tails” last portion of the distillate was discarded. The “hearts” distillate was stored in 750 mL glass bottles at room temperature. It may be diluted with DI water to reach a similar alcohol by volume from sample to sample before a sensory evaluation. 

### 2.5. HPLC Analysis

High-performance liquid chromatography (HPLC) was used to measure the following: maltose, glucose, lactic acid, glycerol, acetic acid, and ethanol in the mash; a pre-fermentation evaluation; and in the distiller’s beer, a post-fermentation evaluation. A 25 mL sample per batch was taken after mashing, and another 25 mL sample was taken after fermentation; the samples were kept at −20 °C. For HPLC evaluation, “mash” and “beer” samples were centrifuged at 4000× *g* with an Eppendorf 5810R centrifuge (Hamburg, Germany). They were filtered to avoid any solids transferring into the HPLC vial. A final solution was made of 1.2 mL containing 1.14 mL of each mash or beer sample and 0.06 mL of DI water. The HPLC procedure was described by Sluiter [22] and adapted by Dodge [23]. Following the two-stage acid hydrolysis, the amounts of glucose and maltose, lactic acid, acetic acid, glycerol, and ethanol were measured using a Dionex UltiMate 3000 HPLC instrument (Dionex Corporation, Sunnyvale, CA, USA) equipped with a refractive index detector and a Biorad Aminex HPX-87H column, using 5 mM H_2_SO_4_ as the mobile phase at a flow rate of 0.4 mL/min and a column temperature of 50 °C (Hercules, CA, USA). The organic solvents, hexane, petroleum ether, chloroform, ethyl acetate, IPA, and palladium on activated charcoal (Pd/C), were purchased from Sigma-Aldrich, St. Louis, MO, USA. 

### 2.6. Data Analysis and Visualization

All values obtained are reported by batch, and the average of the eight batches was calculated to obtain a mean and standard error and is also reported in the Section 3. All analyses were performed and figures were created with R software (R-4.1.2 for Windows), using the package ggplot2.

## 3. Results and Discussion

Before cooking each mash, we evaluated whole corn samples with an NIR grain analyzer. The average and standard deviation for moisture, protein, oil, crude fiber, ashes, and starch are shown in Table 1. Moisture content in the samples ranged from 8.9% to 13%. Protein content ranged from 7.2% to 8.1% with an average of 7.5%, and oil content had an average of 3.76%, ranging from 3.39% to 4.37%. Starch content had an average of 65.9% and ranged from 64.5% to 66.9%. All samples have a grain composition in the range of corn grain obtained by distilleries [24,25], which indicates the quality of the grain used for the development of this methodology for new-make whiskey production.

Our data do not yet allow us to make conclusions about the effect of starch content on distillation parameters. Starch is a key source of fermentable sugars, and its relationship with alcohol yield and other distillation parameters has been studied in wheat [12], but more research is needed to understand this relationship for corn, which is the main component of the mash bill for bourbon whiskey production. It is suggested that metabolites coming from protein and oil degradation contribute to the sensory profile of the final whiskey [26,27,28,29]; therefore, their content should be considered and measured by researchers.

A mash sample of 25 mL was collected after cooking each mash and before adding the yeast, and glucose and maltose, two key saccharides, were quantified with HPLC. Figure 1 shows the glucose and maltose content for each mash. The glucose content ranged from 70 to 98 g/L, and the maltose content ranged from 49.2 to 67 g/L. The sum of both saccharides showed a concentration that ranged between 121.6 g/L and 160 g/L. 

The coefficient of variation (CV) among samples for glucose content was 12.5%; we see this as a “batch effect”. Some factors that could have impacted this are the different sources of corn grain we used to develop and adjust this methodology, room temperature variation during mash cooling down, enzymatic action in the different mashes, and human error. Training the staff or students working on this type of project is key to minimizing human error and achieving consistency between batches as we could observe error if enzyme concentrations varied, temperatures varied, or the duration of process intervals varied.

A “beer” sample of 25 mL was collected after fermentation (96 h) and glucose and maltose, two key saccharides, were quantified with HPLC. The content for each batch is reported in Appendix A. Figure 2 shows the average glucose and maltose content (g/L) evaluated at pre-fermentation (0 h) and at 96 h of fermentation initiated. The reduction in glucose content during fermentation was observed from an average value of 84 to 16 g/L and maltose content from 58 to 2 g/L. The HPLC quantifications showed that the fermentation was accomplished for every sample with a remaining 3% of maltose and 19% of glucose. 

The length of fermentation varies among distillers [30]. We are aware that distilleries are reducing this time to 36–48 h of fermentation and adding specific mixtures of yeast combined with specific temperatures to speed up the process. With a sample of 2.1 kg of milled grain, we achieved the desired levels of fermentable sugars after 4 days (96 h), consistently for all samples. Other methodologies found in the literature established fermentation processes ranging from 68 h [13,14,31] and 72 h [11] to 120 h [9].

Other compounds measured with HPLC to evaluate the performance of the mashing and fermentation processes were glycerol, lactic acid, and acetic acid. When measured on the “mash” samples, there was no presence of these compounds. Table 2 shows the “beer” content for these compounds. Glycerol content ranged from 1.94 to 2.8 g/L, lactic acid ranged from 1.18 to 4.82 g/L, and acetic acid ranged from 0.3 to 0.9 g/L. These values are in line with what is desired by the industry [32,33], and the range we measured of lactic and acetic acid content is between the normal values. Lactic acid and acetic acid production should be monitored. Increasing levels of these acids could be due to stress factors during fermentation and will generate lower ethanol yields and/or the end of fermentation activity [33,34].

As stated before, a first distillation was performed with a copper pot still. “Low wines” were collected, and ethanol was measured with a hydrometer for each sample. Figure 3 shows the increase in ethanol concentration measured in %ABV (percentage of alcohol by volume) at the three time points: 96 h (end of fermentation), on low wines (after first distillation), and after a second distillation. The ethanol content on average between the eight samples ranged from 10% ABV initially after fermentation to 77% ABV after a second distillation. The variation between samples was very small as can be seen by the standard error. The percentage of alcohol by volume obtained at each step is in line with the ranges expected by the industry for each process [35]. 

Increasing alcohol yield is a primary objective of distilleries, and the efficiency reached in the conversion of starch into ethanol is a major determinant of this. The cost of the raw material is a major component of the overall costs of running a distillery [36]. The association between starch content and alcohol yield has been studied in wheat, maize, and sorghum [10]. The relationship between the nitrogen (N) content of the grain and alcohol yield has been very well studied in wheat [10,12,14,37], and a negative association between alcohol yield and protein content has been found, but there is no such clear conclusion in maize. The size to which the maize grain is ground, called particle size, for mashing is another topic that has been discussed [25,38], with some distillers aiming for fine particles (2 mm) like the ones described in research articles and others preferring bigger particle sizes. Research is still needed to elucidate these aspects of corn grain influencing different processes of bourbon whiskey production, and this lab-scale protocol aims to provide researchers in this field a methodology to leverage corn research for distilling quality and breakthrough technologies to bring innovation to the bourbon distillery industry.

Figure 4 shows the equipment set up to accomplish the mashing, fermentation, and distillation processes.

This setup can be installed in any laboratory, and our data showed consistency between batches, ensuring the credibility and reproducibility of the results. While this methodology focuses on corn, other grains could also be used similarly with mild alterations to the method. These alterations would be related to the gelatinization, scarification, and liquidation temperatures and hold times, as other grains generally do not need the same extraction conditions as corn. Furthermore, using these conditions on another grain such as rye or wheat would not accurately represent what occurs in the distillation industry. When adjusting the grain or other conditions, two areas should be focused on: the first is ensuring that the mash will be fermented to a standardized amount; the second is that, before the second distillation, each sample should be standardized to the same alcohol content. These factors are important to minimize possible variation, as an alteration in alcohol content can change the volatility properties of compounds. The distillate produced in this method requires two passes to produce approximately 500 mL of distillate, which can be used for sensory analysis, or the purposes defined by researchers. The two passes are required to remove unwanted compounds from the distillate that will impact sensory attributes. The volume-based cuts depend upon experimental conditions and may be altered depending on experimental objectives. 

This method could be improved to produce a greater volume of distillate and better reproducibility with the addition of specialized equipment to measure smaller aliquots of ethanol content, and therefore to reach more precise cuts during both distillation steps. 

## 4. Conclusions

This study presents a lab-scale methodology for new-make bourbon whisky production. Our aim was to develop a methodology that allows us to use small quantities of grain (~2.5 kg), which opens new avenues for future analysis of different hybrids or varieties, even breeding lines for distilling quality performance or the effect of terroir with simplified handling of the grain. We achieved a second objective, to perform each part of the process separately with specific time points to quantify sugars, alcohol, and other primary metabolites. We accomplish our third goal of producing around 500 mL of final distillate for further sensory profiling, including tasting. This micro-distillation method provides a robust tool for continuous improvement in the bourbon whiskey category. 

## Figures and Tables

**Figure 1 foods-12-00457-f001:**
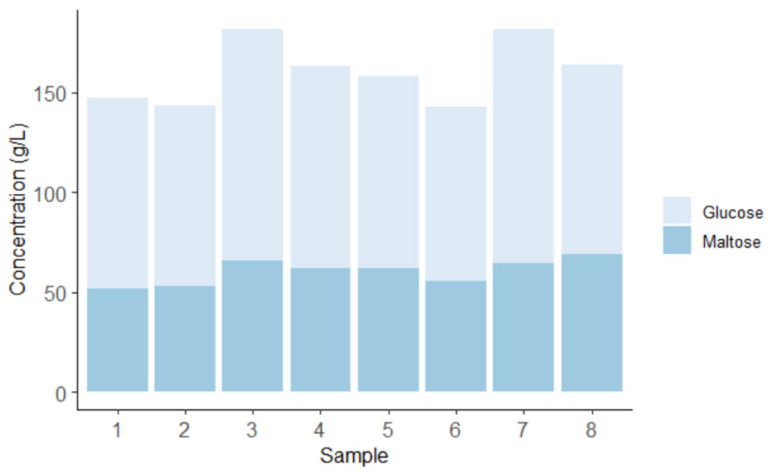
Analysis of the mash: pre-fermentation saccharide content. Each bar represents an individual sample.

**Figure 2 foods-12-00457-f002:**
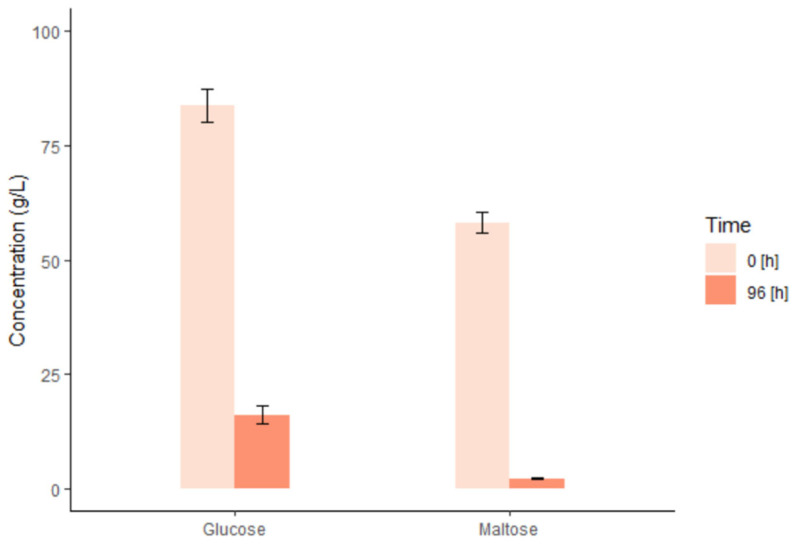
Mean glucose and maltose content and standard error (SE) pre (0 h) and post-fermentation (96 h). *N* = 8.

**Figure 3 foods-12-00457-f003:**
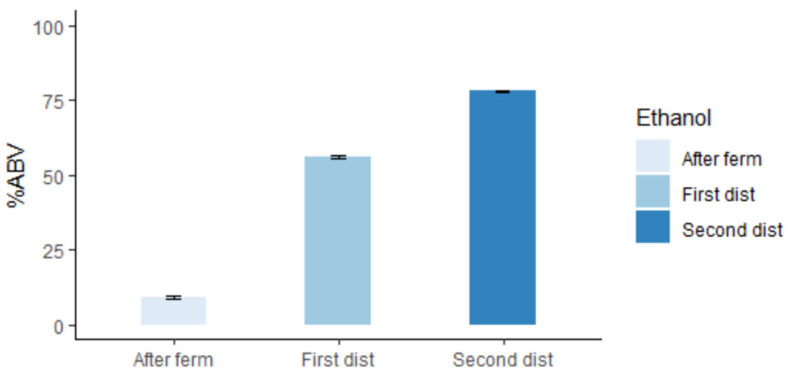
Mean ethanol content and standard error (SE) at each step for the obtention of new-make bourbon whisky. %ABV: percentage alcohol by volume, After ferm: 96 h after fermentation initiated, First dist: ethanol content of “low wines”, Second dist: ethanol content of “hearts”, distillate obtained after a second distillation.

**Figure 4 foods-12-00457-f004:**
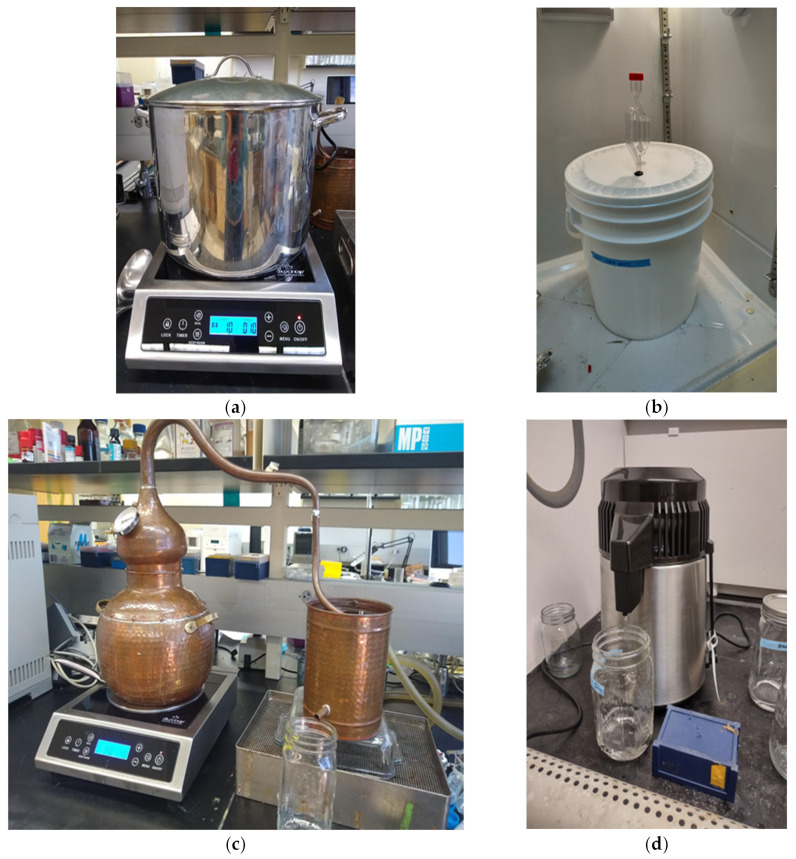
Laboratory equipment for new-make bourbon whiskey production: (**a**) mashing; (**b**) fermentation; (**c**) first-pass distillation; (**d**) second-pass distillation.

**Table 1 foods-12-00457-t001:** Average and standard deviation for moisture content (%), protein content (%), oil content (%), crude fiber content (%), ash content (%), and starch content (%) for each sample (*N* = 2) and the average of the eight samples. *N* = 8.

Sample	Moisture %	Protein %	Oil %	C. fiber %	Ash %	Starch %
1	11.98 ± 0.06	7.27 ± 0.08	3.39 ± 0.05	1.70 ± 0.01	1.07 ± 0.01	65.21 ± 0.13
2	10.51 ± 0.17	7.57 ± 0.11	3.99 ± 0.51	1.66 ± 0.08	1.20 ± 0.02	66.61 ± 0.09
3	10.78 ± 0.18	7.23 ± 0.06	4.37 ± 0.41	2.68 ± 0.03	0.86 ± 0.06	65.85 ± 0.35
4	8.89 ± 0.17	8.10 ± 0.11	3.60 ± 0.03	1.91 ± 0.05	1.26 ± 0.01	66.26 ± 0.25
5	9.62 ± 0.01	7.39 ± 0.17	3.52 ± 0.17	1.93 ± 0.06	1.24 ± 0.01	66.93 ± 0.50
6	9.75 ± 0.03	7.61 ± 0.15	3.56 ± 0.15	2.02 ± 0.09	1.24 ± 0.01	66.02 ± 0.24
7	13.12 ± 0.16	7.30 ± 0.16	3.64 ± 0.16	1.58 ± 0.06	1.26 ± 0.03	65.40 ± 0.26
8	12.20 ± 0.27	7.68 ± 0.01	4.06 ± 0.01	1.57 ± 0.03	1.18 ± 0.06	64.54 ± 0.30
Average	10.87 ± 1.48	7.52 ± 0.29	3.76 ± 0.34	1.88 ± 0.36	1.16 ± 0.14	65.98 ± 0.89

**Table 2 foods-12-00457-t002:** Glycerol, acetic acid, and lactic acid content after fermentation (96 h) by sample and mean and SE among the 8 samples.

Sample	Glycerol (g/L)	Acetic Acid (g/L)	Lactic Acid (g/L)
1	2.80	0.42	3.66
2	2.12	0.34	1.18
3	2.62	0.42	4.34
4	1.96	0.94	4.82
5	2.84	0.94	2.48
6	2.00	0.70	2.88
7	2.26	0.92	2.46
8	1.94	0.40	3.52
Average	2.32 ± 0.38	0.64 ± 0.27	3.17 ± 1.16

## Data Availability

The data analyzed and generated in this study are available from the corresponding author upon request.

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
