# Peer review of "Lab-Scale Methodology for New-Make Bourbon Whiskey Production"

_foods, 2023, doi:10.3390/foods12030457_

Round 1

Reviewer 1 Report

Comments to the authors:

Bourbon whisky has a long tradition in the USA. A scientific, professional approach to the production and quality control of these is essential. The development of a laboratory-based methodology to help the next generation learn how to make and control the quality of bourbon whisky is to be welcomed.

The authors have produced a sufficiently detailed methodology that can be implemented in the laboratory.

The authors have duly contextualised their objective. The description of the methodological part is sufficiently thorough and reproducible. The authors have reviewed both previous and recent relevant literature and do not include self-citation. The manuscript is well structured.

However, please provide some details:

·       The authors did not explain why they tested maize samples from different origins. Which of the eight samples can be considered a standard?

·       Table 1 does not indicate the origin of the samples. If the origin of the samples is not considered as a relevant parameter, please explain why.

·       Why are the values obtained by NIR determination averaged over the eight samples. Are the parameters of each sample not relevant or is there no significant difference between them?

·       Can the values obtained in Table 1 be used to categorise the samples qualitatively? Possibly make a recommendation as to the values at which the maize is suitable for producing quality bourbon whisky?

·       Is it not clear whether the data presented in Tables 1 to 2 and Figures 1 to 3 are acceptable? For the beginner or non-expert, it is important to know within which limits the parameters to be measured at each step of the whisky production process are acceptable. These parameters are the basis for judging whether the fermentation has been carried out correctly. The authors should take a position on whether their data are indicative. I believe that this is to be expected when developing a laboratory protocol.

·       Figure 2 shows the average values for the eight samples. Justification is given as to why the variation in glucose and maltose content is not presented on a sample by sample basis.

I would like to draw attention to a few formal shortcomings:

·       In line 102, I would suggest writing "near-infrared (NIR) spectrometric determination".

·       In line 109, replace "lt" with the official SI notation as "L" if this is used elsewhere in the text. In line 128, “gr” is replaced by 'g' to denote grams. The use of "ml" and "mL" is not consistent. Use “h” for the hour, not 'hr'.

·       There are numerous absences in the text of end of sentence periods, spaces between words, and in several places “C” instead of “OC” for Celsius degree.

·       The sample number is omitted from the explanation of Figure 3.

·       Figure 4 requires some editing.

·       The editing of the Reference chapter is not consistent.

·       The availability of the data has not been declared by the authors, please supply this information.

Author Response

Dear Reviewer, please, see the attachment.

Reviewer 2 Report

Dear authors the paper illustrates a Lab-scale Methodology to produce  Bourbon Whiskey in a new way. It is an interesting method also useful for students.

It needs some minor revisions of the English language

line 12: had (remove has)

line18:  a copper pot still pot ? maybe a copper pot still 

line 28: 160 proof and 125 proof are 160° proof ...125° proof

line 203-208: it is better to put this part in introduction and in materials and methods. Moreover the English style can be improved. 

line 222: Some factors that could have an incidence in this are..

better Some factors that could impact this are......

line 230-232: The reduction in glucose content during fermentation can be seen; on average went from 84 to 16 gr/L and the maltose content decreased  from 58 to 2 gr/L.

better The reduction of glucose content during fermentation was observed from an average value of 84 to 16 g/L and of maltose from 58 to 2 g/L.

line 240: How long fermentation should last varies among distillers better

The length of fermentation varies among distillers

line 243-244: what do you mean with consistently among samples ?

Author Response

(The authors gave the same response as above.)
